

# The potential effects of climate change on air quality across the conterminous U.S. at 2030 under three Representative Concentration Pathways (RCPs)

Christopher G. Nolte[1], Tanya L. Spero[1], Jared H. Bowden[2], Megan S. Mallard[1], and Patrick D. Dolwick[3]

[1]Office of Research and Development, U.S. Environmental Protection Agency, Research Triangle Park, North Carolina, USA
[2]North Carolina State University, Raleigh, North Carolina, USA
[3]Office of Air Quality Planning and Standards, U.S. Environmental Protection Agency, Research Triangle Park, North Carolina, USA

**Correspondence:** C. G. Nolte (nolte.chris@epa.gov)

**Abstract.** The potential impacts of climate change on regional ozone ($O_3$) and fine particulate ($PM_{2.5}$) air quality in the United States are investigated by downscaling Community Earth System Model (CESM) global climate simulations with the Weather Research and Forecasting (WRF) model, then using the downscaled meteorological fields with the Community Multiscale Air Quality (CMAQ) model. Regional climate and air quality changes between 2000 and 2030 under three Representative Concentration Pathways (RCPs) are simulated using 11-year time slices from CESM. The regional climate fields represent historical daily maximum and daily minimum temperatures well, with mean biases less than 2 K for most regions of the U.S. and most seasons of the year and good representation of variability. Precipitation in the central and eastern U.S. is well simulated for the historical period, with seasonal and annual biases generally less than 25%, with positive biases exceeding 25% in the western U.S. throughout the year and in part of the eastern U.S. during summer. Maximum daily 8-h ozone (MDA8 $O_3$) is projected to increase during summer and autumn in the central and eastern U.S. The increase in summer mean MDA8 $O_3$ is largest under RCP8.5, exceeding 4 ppb in some locations, with smaller seasonal mean increases of up to 2 ppb simulated during autumn and changes during spring generally less than 1 ppb. Increases are magnified at the upper end of the $O_3$ distribution, particularly where projected increases in temperature are greater. Annual average $PM_{2.5}$ concentration changes range from $-1.0$ to $1.0\,\mu g\,m^{-3}$. Organic $PM_{2.5}$ concentrations increase during summer and autumn due to increased biogenic emissions. Aerosol nitrate decreases during winter, accompanied by lesser decreases in ammonium and sulfate, due to warmer temperatures causing increased partitioning to the gas phase. Among meteorological factors examined to account for modeled changes in pollution, temperature and isoprene emissions are found to have the largest changes and the greatest impact on $O_3$ concentrations.

## 1 Introduction

In the United States, emissions that lead to the formation of ozone ($O_3$) and atmospheric particulate matter (PM) have declined significantly in recent decades, resulting in substantial improvements in air quality (Parrish et al., 2011; U.S. EPA, 2012) and consequent benefits for human health (Pope III, 2007; Correia et al., 2013). As a result of regulatory actions, such as the



Cross-State Air Pollution Rule and the Tier 2 and Tier 3 emissions standards for motor vehicles, anthropogenic emissions are projected to continue their downward trend through 2030 (U.S. EPA, 2015), leading to further reductions in ambient $O_3$ and concentrations of PM particles smaller than 2.5 μm in diameter ($PM_{2.5}$).

Because air pollution is highly sensitive to meteorology, climate change has the potential to affect air quality by modifying temperatures, wind speeds, mixing heights, humidity, clouds, and precipitation, which all affect pollutant formation and removal rates (Jacob and Winner, 2009; Tai et al., 2010; Fiore et al., 2015; Westervelt et al., 2016). Studies using global climate model (GCM) data to drive global or regional chemical transport models (CTMs) have found that climate change yields meteorological conditions that are more conducive to forming high $O_3$, exacerbating summertime $O_3$ over polluted continental regions (Mickley et al., 2004; Leung and Gustafson, 2005; Stevenson et al., 2006; Katragkou et al., 2011; Horton et al., 2012; Gao et al., 2013). Modeling studies conducted using mid-21st century climate data project up to 2–8 ppb increases in summer average ozone levels in the United States, depending on climate change scenario and time period (e.g., Wu et al., 2008; Nolte et al., 2008; Weaver et al., 2009; Kelly et al., 2012; Trail et al., 2014; Pfister et al., 2014; Gonzalez-Abraham et al., 2015; Fann et al., 2015; He et al., 2016; Dionisio et al., 2017). This deterioration of air quality due to climate change is known as the "climate penalty" (Wu et al., 2008; Rasmussen et al., 2013) and could potentially offset some of the improvement in air quality that would otherwise occur due to reductions in ozone precursor emissions. The strong evidence for the increase in surface $O_3$ levels due to climate change was cited in support of the finding that emissions of greenhouse gases (GHGs) endanger human health and welfare and are therefore subject to regulation in the U.S. under the Clean Air Act (U.S. EPA, 2009). The net effects of climate change on $PM_{2.5}$ are more uncertain. Some studies that have investigated the impacts of climate change on $PM_{2.5}$ have found small but statistically significant effects of 0.5–2.0 $\mu g\, m^{-3}$, but with little consistency, even in the sign of the change (Liao et al., 2006; Racherla and Adams, 2006; Heald et al., 2008; Pye et al., 2009; Kelly et al., 2012; Tai et al., 2012; Dawson et al., 2014; Day and Pandis, 2015; Fiore et al., 2015; Shen et al., 2017). It should be noted, however, that most studies of climate change impacts on $PM_{2.5}$ have neglected changes in climate-sensitive PM emissions sources. The studies that have considered changes in these sources have concluded that warmer temperatures and earlier snowmelt associated with climate change will lead to increased impacts from wildfires (Spracklen et al., 2009; Val Martin et al., 2015; Liu et al., 2016) and dust storms (Achakulwisut et al., 2018).

Motivated by high positive biases (exceeding 10 ppb) in present-day $O_3$ obtained in previous work (Nolte et al., 2008), which were attributed to positive biases in temperature in the downscaled meteorology, we developed improved regional climate modeling techniques that were tested by downscaling coarse reanalysis data (Bowden et al., 2012; Otte et al., 2012; Bowden et al., 2013) and using the downscaled meteorology to simulate air quality (Seltzer et al., 2016). In the present study, we apply this downscaling methodology to GCM data and use the resulting regional climate fields to drive simulations of air quality across the conterminous U.S. The near-future timeframe of 2030 is chosen because of its relevance for air quality policy and the current planning horizon. Lateral boundary conditions and anthropogenic emissions are identical for both the historical and future periods to isolate the meteorological influences of near-term climate change on regional air quality. The simulated historical regional climate is evaluated by comparison to reanalysis fields. Changes in regional climate and air quality at 2030 are presented, and we relate the changes in air quality to the meteorological drivers for these changes.



Previous studies of the effects of climate change on air quality have typically considered a single climate scenario (Trail et al., 2014; Gonzalez-Abraham et al., 2015), a few-year period (Nolte et al., 2008; Gao et al., 2013; Penrod et al., 2014) or single season (Kelly et al., 2012; Pfister et al., 2014; Day and Pandis, 2015; Gonzalez-Abraham et al., 2015). This study examines the impact of climate change on both ozone and PM air quality for the full annual cycle using 11-year periods with three

GHG trajectories. In addition to presenting changes in seasonal mean quantities, we also focus on distributions and examine variability across seasonal and diurnal temporal scales.

## 2   Modeling Approach

### 2.1   Global Climate Model

The GCM used in this study is the National Center for Atmospheric Research-Department of Energy Community Earth System

Model (CESM) (Gent et al., 2011). The model has horizontal grid spacing of $0.9°$ latitude $\times$ $1.25°$ longitude. Eleven-year time slices from simulations conducted for the fifth phase of the Coupled Model Intercomparison Project (CMIP5) (Taylor et al., 2012) were selected for downscaling: 1995–2005 at the end of the CMIP5 historical 20th century simulation, as well as 2025–2035 from simulations following three Representative Concentration Pathways (RCPs; van Vuuren et al., 2011). The RCP8.5 scenario (Riahi et al., 2011) assumes "business as usual," where GHG concentrations increase substantially over the 21st

century, leading to $8.5 \ \mathrm{W \ m^{-2}}$ radiative forcing by 2100. The RCP6.0 scenario (Masui et al., 2011) assumes a modest degree of mitigation of GHG emissions, where total radiative forcing increases before stabilizing at $6.0 \ \mathrm{W \ m^{-2}}$ in 2100. The RCP4.5 scenario (Thomson et al., 2011) has a GHG emissions peak in the middle of the 21st century followed by a decline, so that total radiative forcing is $4.5 \ \mathrm{W \ m^{-2}}$ in 2100. Although the RCP scenarios are named for their radiative forcing at the year 2100, the GHG emissions paths in each scenario were developed by independent modeling groups. As a result, a lower RCP

scenario may have higher GHG emissions and a greater increase in global average temperature than a higher RCP scenario for the 2025–2035 period examined here (Collins et al., 2013).

### 2.2   Regional Climate Model

The CESM data were downscaled with the Weather Research and Forecasting (WRF) model (Skamarock and Klemp, 2008) version 3.4.1 to a domain with 36-km horizontal grid spacing covering most of North America ($199 \times 127$ grid points; Fig. 1)

and 34 vertical layers extending to a model top at 50 hPa. Archived 6-h fields used for downscaling included 3-D temperature, specific humidity, horizontal wind components, pressure, and geopotential height; 2-D surface pressure, skin temperature, 2-m temperature, and 2-m specific humidity; as well as monthly average sea surface temperatures, ice fraction, soil moisture, and soil temperature. To avoid water temperature discontinuities that arise from applying GCM ocean temperatures to large lakes (Mallard et al., 2015), monthly lake temperature data from the land component of CESM (i.e., the Community Land Model,

CLM) were used to set the temperature of inland water points on the regional domain (Spero et al., 2016). All monthly fields were temporally interpolated to 6-h intervals to avoid abrupt transitions in the regional climate simulations.



WRF was initialized at 0000 UTC 1 October 1994 for the historical run and at 0000 UTC 1 October 2024 for each of the RCP runs, so that each regional climate simulation included a 3-month spin-up period. Land use classification was based on the 24-category USGS land cover database. WRF was configured as in Spero et al. (2016), with spectral nudging of horizontal wind components, potential temperature, and geopotential applied above the planetary boundary layer (PBL) using the nudging
coefficients from Otte et al. (2012).

## 2.3  Chemical Transport Model

The chemical transport model used was the Community Multiscale Air Quality (CMAQ) model (www.epa.gov/cmaq) version 5.0.2 (Byun and Schere, 2006; Appel et al., 2013; U.S. EPA, 2014a). The model was configured with the multipollutant version of the Carbon Bond 2005 gas phase chemical mechanism (cb05tump) and the AERO6 aerosol module (Simon and Bhave,
2012; Nolte et al., 2015). CMAQ simulations were conducted over a 36-km domain covering the conterminous U.S. (148 × 110 grid cells; Fig. 1). The Meteorology-Chemistry Interface Processor (MCIP) (Otte and Pleim, 2010) version 4.1.3 was used to prepare meteorological fields for CMAQ using the same vertical layering as in WRF. Reported pollutant concentrations are taken from the lowest model layer, which has a depth of about 38 m. Each 11-year CMAQ simulation was run continuously following a 10-day spin-up period.

Numerous studies using regional CTMs that have considered both changing climate and changing emissions on future air pollutant concentrations have found that changes in emissions dominate (Nolte et al., 2008; Kelly et al., 2012; Colette et al., 2013; Trail et al., 2014; Day and Pandis, 2015; Gonzalez-Abraham et al., 2015; He et al., 2016). Modeled pollutant concentrations are highly sensitive to lateral chemical boundary conditions (e.g., Tang et al., 2007; Katragkou et al., 2010; Schere et al., 2012), and different assumptions regarding changes in long-range transport have been shown to have a significant im-
pact on future pollutant levels (Nolte et al., 2008; Colette et al., 2013; Pfister et al., 2014; Gonzalez-Abraham et al., 2015; He et al., 2016; Zhang et al., 2016). Several previous studies have also highlighted the importance of rising levels of methane for ozone chemistry (Fiore et al., 2002; West and Fiore, 2005; Nolte et al., 2008). To isolate the effects of climate change on air quality, only the meteorological conditions and the meteorologically-dependent emissions that are modeled within CMAQ were modified between the historical and future CMAQ simulations. All other input variables, including anthropogenic emis-
sions, chemical lateral boundary conditions, and land use and land cover classifications, were unchanged across the air quality modeling scenarios.

Chemical lateral boundary conditions were derived from an independent simulation of the year 2011 by the GEOS-Chem global chemical transport model (Bey et al., 2001; Henderson et al., 2014) and were used for each year of the historical and the RCP simulations. Anthropogenic emissions for each year of both the historical and future periods were modeled using
the 2030 emissions projection that was used as the reference case for the Tier 3 motor vehicle standards rulemaking analyses (U.S. EPA, 2014b, c). This projection assumed the implementation of previously adopted air quality policies, with the result that anthropogenic $NO_x$, $SO_2$, and volatile organic compound (VOC) emissions are 54%, 69%, and 25% lower, respectively, than in the 2011 National Emissions Inventory (Table S1). Biogenic VOC emissions were allowed to vary according to climate-



driven meteorological changes. Monthly and diurnal temporal profiles were applied to other emissions source sectors, including wildfires, but did not vary across years. Emissions of $NO_x$ due to lightning were not modeled.

## 3    Evaluation for Historical Period

The CMAQ modeling system has been extensively evaluated for simulation of historical ("retrospective") air quality (Foley et al., 2010; Appel et al., 2013, 2017). It is challenging, however, to evaluate air quality simulated using meteorology downscaled from a global climate model (Menut et al., 2013; Seltzer et al., 2016). Because climate models are run without assimilating weather observations, the weather conditions simulated by downscaling a GCM for a particular historical day cannot be expected to correspond to the hourly meteorology that occurred on that day. For the same reason, it is inappropriate to evaluate air pollutant concentrations simulated using downscaled meteorology against hourly or daily historical measurements. Instead, regional climate and air quality should be evaluated at seasonal and monthly temporal scales. As a further complication, to account for interannual meteorological variability it is necessary to run the model for periods of several years or even decades, but anthropogenic emissions of pollutants such as $NO_x$, VOC, and $SO_2$ can exhibit significant trends that confound the analysis of the impact of using downscaled meteorology. An evaluation of 2000–2010 ozone and $PM_{2.5}$ air quality simulated using historical emissions and meteorology downscaled from a coarse-scale historical reanalysis showed comparable performance compared with typical air quality modeling applications (Seltzer et al., 2016). This demonstrates that the downscaling procedure does not introduce substantial bias into the modeled air quality, providing confidence in the method's use for future air quality projections.

Because this study uses projected 2030 emissions in all simulations, including for the historical period, modeled air quality is not compared to observations. Instead the evaluation of the historical period is focused on monthly and seasonal means and selected percentiles of regional temperature and precipitation, two meteorological fields that strongly affect air quality. Two-meter temperature and precipitation from the historical period are evaluated by comparing against the Climate Forecast System Reanalysis (CFSR; Saha et al., 2010) and the North American Regional Reanalysis (NARR; Mesinger et al., 2006), respectively. CFSR is a global reanalysis with hourly 2-m temperature at $0.31°$ resolution, enabling evaluation of daily maximum and daily minimum temperatures. NARR is used to evaluate precipitation because it has been shown to represent precipitation well over the conterminous U.S. (Bukovsky and Karoly, 2007), while the CFSR has a wet bias (Otte et al., 2012). Regional analysis is performed using U.S. climate regions defined by the National Centers for Environmental Information (Fig. 1).

Seasonal averages of daily mean 2-m temperatures simulated by downscaling CESM with WRF are compared against CFSR fields horizontally interpolated to the WRF grid in Fig. 2. The seasonal and spatial patterns of 2-m temperatures are generally well represented by WRF, though in areas of complex terrain in the western U.S. there are positive and negative biases exceeding 4 K (Fig. 2). Daily minimum temperatures are within $\pm$ 2 K of CFSR for every region and season except during summer (JJA) in the Northwest and West regions, which have warm biases of 2.7 and 3.4 K, respectively. Daily maximum temperatures are also generally well simulated, with absolute biases exceeding 2 K only for the Southwest during spring (MAM), the Upper Midwest during spring, summer, and autumn (SON), and for the Northeast during spring and autumn (Fig. 3). Though these





regionally averaged temperature biases are somewhat larger than typically obtained in retrospective meteorological modeling for air quality applications, they are comparable to biases reported in dynamically downscaled meteorology utilizing nudging (e.g., Trail et al., 2013; Gonzalez-Abraham et al., 2015; Colette et al., 2013). We note that these biases in the downscaled regional climate fields are largely attributable to the driving CESM fields rather than to errors within WRF (see Supplementary

Information (SI)).

Distributions of the daily maximum 2-m temperatures simulated by WRF for each region and month over the historical 1995–2005 period in comparison to CFSR are shown in Fig. 3; regional distributions of daily minimum 2-m temperatures are provided in the SI. The downscaled simulations using WRF reasonably capture the regional variation in the annual cycle of median values as well as the width of the interquartile range (IQR). Narrower distributions are simulated during summer than

winter, in agreement with the pattern in CFSR, but WRF accentuates this difference in some regions, with excessively narrow daily maximum temperature distributions simulated in the Upper Midwest, Northeast, Ohio Valley, South, and Southeast. For maximum temperatures, the WRF simulations of the Northwest and the Northern Rockies regions have the best overall agreement with CFSR. Though maximum temperatures are negatively biased most of the year in the Southwest and West, the magnitude of the IQR is well represented in those regions. During the summer, the IQR of daily maximum temperatures in

WRF is much lower than in CFSR in several regions, including the Upper Midwest, South, Ohio Valley, and Southeast. In the regions and months with the largest biases, the distribution is shifted by nearly a quartile. The worst performance is in August in the Upper Midwest, in which the 25th and 50th percentile daily maximum temperatures simulated by WRF exceed the median and 75th percentile CFSR values, respectively.

The spatial and seasonal distributions of precipitation across the conterminous U.S. in WRF are broadly consistent with

NARR (Fig. 2). WRF generally has a wet bias relative to NARR, except for the South, Upper Midwest, and Ohio Valley regions. Regional biases relative to NARR are given in Table 1. Precipitation is reasonably well simulated in the central and eastern U.S., with most seasonal and annual biases 25% or less. In the western U.S., however, WRF precipitation is positively biased relative to NARR throughout the year, particularly in the Southwest during winter and spring and in the West and Northwest regions during summer. A less severe positive bias in precipitation also exists during the summer in the eastern U.S.

north of Florida.

## 4   Changes at 2030 Under RCPs

Potential changes in seasonal mean air pollutant concentrations are presented under the three RCPs for 2025–2035 relative to 1995–2005. Next, the meteorological drivers influencing the changes in air quality are examined.

### 4.1   Ozone

Changes in seasonal mean maximum daily 8-h average (MDA8) $O_3$ levels for spring, summer, and autumn are shown in Fig. 4; plots showing absolute magnitudes are provided in the SI. The general locations of the seasonal changes are consistent across the three RCP scenarios, although the magnitudes are less pronounced under RCP4.5 and RCP6.0, as expected. Statistically



significant increases of 1–5 ppb are simulated during summer under RCP8.5 across most of the northern and eastern U.S., with regional average increases of at least 2 ppb across the Northern Rockies, Upper Midwest, and Ohio Valley (Table 2). Summer decreases of up to 1.5 ppb are projected in the South and Southeast regions, particularly along the Texas Gulf Coast. The summer decrease is widespread under RCP4.5 and RCP6.0, averaging 0.4–0.5 ppb across the South and Southeast regions.

The projected impact of climate change on MDA8 $O_3$ is lower during the spring and autumn seasons than in summer under all three RCPs. For the spring, small increases of 0.5–1.0 ppb are simulated over parts of the Ohio Valley, South, and Upper Midwest regions under RCP4.5 and RCP8.5, which generally are not statistically significant. Statistically significant decreases of 0.5–1.0 ppb are simulated along the Southeast coast in RCP6.0. During the autumn, significant increases of 1–2 ppb are simulated across a broad area of the central U.S. including most of the South, Ohio Valley, and Upper Midwest regions under

both RCP8.5 and RCP4.5, but no significant change is evident under RCP6.0.

The preceding analysis focused on changes in seasonal mean MDA8 $O_3$. Because compliance with the U.S. National Ambient Air Quality Standard (NAAQS) for $O_3$ is assessed using the annual 4th-highest MDA8 $O_3$ ("HI4"), changes in HI4 averaged across the 11-year periods are also shown in Fig. 4 and Table 2. Under RCP8.5, regional average increases in HI4 exceeding 3 ppb are simulated for the Upper Midwest, Ohio Valley, and Northeast, with increases exceeding 5 ppb over large

areas within those regions as well as parts of the Southwest and West. Under RCP4.5 and RCP6.0, regional average HI4 increases 1.0–1.7 ppb in the Upper Midwest, Ohio Valley, and Northeast, exceeding 3 ppb through large parts of those regions. The modeled increases in HI4 under all three RCPs examined in the Upper Midwest, Ohio Valley, and Northeast regions, which are highly populated areas of the U.S., have potentially significant implications for human health and NAAQS compliance.

Some previous observational (Porter et al., 2015) and modeling studies (Weaver et al., 2009; Jacob and Winner, 2009;

Pfister et al., 2014; Rieder et al., 2015) have found that extreme $O_3$ values have a greater sensitivity to temperature than do mean values. Projected changes across the $O_3$ distribution are examined using seasonal percentiles that are calculated for each grid cell, then averaged across regions and years for the historical and future climate periods (Fig. 5). In summer in the Northeast, Ohio Valley, and Southeast, the change in $O_3$ under each of the RCPs is projected to be greater at the upper end of the distribution. In the South and Southeast, there is a projected decrease of 0.5–1.0 ppb at the lower end of the distribution under

all three RCPs. A gradual increase is projected in the slope through the 90th percentile, with more pronounced increases at the upper tail. By contrast, the projected change in $O_3$ is comparatively uniform across the distribution in the Northern Rockies region, while there is little change at any part of the distribution in the Northwest, West, and Southwest regions. During autumn under RCP8.5, increases ranging from 1 ppb at the low end of the distribution to 2 ppb at the high end are projected in the Upper Midwest and Ohio Valley regions, while changes under RCP4.5 and RCP6.0, as well as changes during spring under

each of the RCPs, are less than 1 ppb throughout the distribution for each region (SI).

To investigate changes over the entire annual cycle, regional monthly boxplots of MDA8 $O_3$ simulated for the historical period and the RCP8.5 simulation are compared in Fig. 6. Analogous comparisons with the RCP4.5 and RCP6.0 runs are included in the SI. Consistent with Fig. 5, the largest changes in median values are projected in the Upper Midwest, Northern Rockies, Northeast, and Ohio Valley regions during the summer. Though some of the highest extreme MDA8 $O_3$ values are

simulated in the West and Southwest, changes in those regions are comparatively small. While most previous studies of the





effect of climate change on $O_3$ pollution have emphasized the summer, when $O_3$ concentrations are highest, a few investigators have reported increases during spring and autumn, suggesting a lengthening of the ozone season (Fiore et al., 2002; Nolte et al., 2008; Trail et al., 2014). Clifton et al. (2014) have projected a reversal of the $O_3$ seasonal cycle in the northeastern U.S. by the end of the 21st century, with increased methane levels and decreased $NO_x$ levels combining to produce a wintertime maximum in surface $O_3$. Here we find that median, 75th, and 98th percentile MDA8 $O_3$ values increase in nearly every region of the U.S. during October, November, and December under the RCPs, but do not show a consistent response during the months January through May.

## 4.2 Particulate Matter

Projected changes in annual mean concentrations of total $PM_{2.5}$ and its largest components under the three RCPs are shown in Fig. 7, while absolute quantities for the historical period and relative changes are provided in the SI. Statistically significant $PM_{2.5}$ decreases of up to $0.7\,\mu g\,m^{-3}$ (5–10%) are simulated in the Northern Rockies and Ohio Valley regions under RCP8.5 and RCP4.5, while increases of up to $1.0\,\mu g\,m^{-3}$ occur in the Southeast under RCP8.5 and RCP6.0. Most of the decreases in $PM_{2.5}$ are due to decreases in nitrate ($NO_3^-$) of up to $0.4\,\mu g\,m^{-3}$ (40%). The decreases in $NO_3^-$ are accompanied by lesser decreases in ammonium ($NH_4^+$) and sulfate ($SO_4^{2-}$). Increases in $PM_{2.5}$ in the Southeast are largely attributable to organic matter (OM), which increases up to $0.5\,\mu g\,m^{-3}$ (10–20%).

Seasonally averaged changes in $NO_3^-$ and OM are shown in Figs. 8–9; the patterns of seasonal changes in $SO_4^{2-}$ and $NH_4^+$ (SI) are similar to the changes in $NO_3^-$. The decreases in annual average $NO_3^-$ levels under RCP8.5 and RCP4.5 (Fig. 7) are driven by decreases during winter and spring (Fig. 8). The decrease is strongest during winter under RCP8.5, when average $NO_3^-$ concentrations decrease by $0.3–0.9\,\mu g\,m^{-3}$ over most of the eastern U.S. By contrast, the increases in OM primarily occur during summer and autumn (Fig. 9). During summer under RCP8.5, projected changes to OM are most pronounced in the Southeast and Ohio Valley regions, where there are projected increases of $0.2–0.8\,\mu g\,m^{-3}$. There are less pronounced increases of $0.1–0.3\,\mu g\,m^{-3}$ in those regions under RCP4.5 and RCP6.0.

## 4.3 Meteorological Influences on Projected Changes in Air Quality

Because the anthropogenic emissions and chemical lateral boundary conditions are the same in all CMAQ simulations, all projected changes in air quality are due to differences in meteorology downscaled from the climate scenarios. To gain insight into the parameters most strongly influencing the changes in air quality, correlation coefficients were calculated between monthly mean changes in several meteorological variables and changes in pollutant concentrations, focusing on the species and seasons where the impacts of climate change were greatest. Variables examined included daily mean, maximum, and minimum 2-m temperatures, daily mean and daily maximum PBL heights, precipitation, cloud cover, 10-m wind speeds, number of days with stagnant meteorological conditions (Wang and Angell, 1999; Horton et al., 2012), and biogenic isoprene emissions. The variables with the strongest correlations to changes in $O_3$ were daily maximum 2-m temperature, isoprene emissions, and cloud cover, while temperature, isoprene, and stagnation had the strongest correlations with $NO_3^-$ and OM (SI).





Projected changes in seasonally averaged daily maximum 2-m temperatures are shown in Fig. 10. As expected, the temperature increase is greatest under RCP8.5. In RCP8.5, daily maximum temperatures increase by 0.5 K across most of the conterminous U.S. in all seasons, by more than 2 K in the South, Upper Midwest, and Ohio Valley regions during winter, and by more than 3 K in much of the Upper Midwest and Ohio Valley regions during summer. Under RCP4.5, daily maximum
temperatures increase by 0.5–3.0 K in most of the conterminous U.S. throughout the year, with the largest and most widespread increase projected during spring. By contrast, the changes in daily maximum temperatures under RCP6.0 are less pronounced, with summertime increases of 1–3 K over most of the U.S. but little change in the eastern U.S. during autumn, and even slight cooling of 0.5–1.0 K projected in parts of the Southeast and Ohio Valley regions during winter. Across the conterminous U.S., annual average daily maximum temperatures increase by 1.2 K under RCP4.5, 0.7 K under RCP6.0, and 1.7 K under RCP8.5.

The spatial patterns of the mean changes in winter and spring daily maximum temperatures in the RCPs (Fig. 10) correspond to the changes in $NO_3^-$ concentrations (Fig. 8), and monthly variations in $NO_3^-$ and maximum temperatures are strongly negatively correlated (SI). This supports the conclusion that warmer temperatures in a future climate result in increased partitioning of aerosol $NO_3^-$ to gas-phase nitric acid ($HNO_3$) (Pye et al., 2009). Aerosol $NO_3^-$ increases in the portions of the Southeast and the Ohio Valley regions where wintertime daily maximum temperatures decrease slightly under RCP6.0. The patterns of
changes in aerosol $NH_4^+$ concentrations (SI) largely mirror the changes in $NO_3^-$. There is little decrease in aerosol $NO_3^-$ during summer because nitrate exists almost totally in the gas phase during that season.

While anthropogenic emissions are unchanged across these simulations, biogenic emissions of VOCs are modeled within CMAQ and respond to changes in meteorology. Isoprene emissions depend on temperature as well as photosynthetically active radiation, which is attenuated in the presence of clouds. Modeled emissions of biogenic isoprene increase across all future
scenarios, due to both warmer temperatures and decreased cloudiness (Fig. 12). Modeled average annual isoprene emissions over the conterminous U.S. increase by 11%, 8%, and 19%, under RCP4.5, RCP6.0, and RCP8.5, respectively. The increased emissions of isoprene and other biogenic VOCs in the heavily forested Southeast region not only enhance production of $O_3$ (Fig. 4), but also account for most of the increases in OM concentrations (Fig. 9).

Scavenging of soluble aerosols by precipitation is an important removal process for atmospheric particulate matter. Shown
in Fig. 11 are percent changes in seasonal precipitation for each of the RCP scenarios. The decrease in summer and autumn precipitation in the South, Southeast, and Ohio Valley regions under all three climate scenarios may be contributing to increases of OM in those regions. However, comparing the changes in seasonal precipitation to changes in $PM_{2.5}$ indicates that changes in aerosol scavenging of soluble aerosols are not strongly affecting average $PM_{2.5}$ concentrations in these simulations. In particular, precipitation decreases strongly in the central U.S. during the winter under all three RCPs, but wintertime $PM_{2.5}$
concentrations decrease in that region under RCP4.5 and RCP8.5, and are largely unchanged under RCP6.0.

The increment in MDA8 $O_3$ per degree of warming projected during summer and autumn varies regionally (Fig. 13), but there is some consistency in spatial patterns between the RCPs. During summer under RCP4.5 and RCP8.5, $\Delta O_3 / \Delta T$ ranges from 0.5–2.0 ppb $K^{-1}$ over much of the Northern Rockies, Upper Midwest, Ohio Valley, and Northeast regions, and 0.5–1.0 ppb $K^{-1}$ in the South, Southeast, Ohio Valley, and Upper Midwest regions during autumn. The temperature change
projected under RCP6.0 during autumn (Fig. 10) is near zero, which explains the extreme values for $\Delta O_3 / \Delta T$. By contrast,





the proportionality between projected $O_3$ and daily maximum temperature is negative in much of the South and Southeast, particularly under RCP4.5 and RCP6.0. This negative relationship between daily maximum temperature and MDA8 $O_3$ is consistent with observation-based sensitivities reported for the Southeast during summer (Camalier et al., 2007; Porter et al., 2015).

**5  Conclusions**

This study investigated the impacts of climate change on regional ozone and $PM_{2.5}$ air quality across the conterminous United States. Eleven-year time slices from global CESM simulations were dynamically downscaled to 36-km horizontal grid spacing with WRF, and these meteorological fields were used by CMAQ to simulate air quality. The climate scenarios represent the year 2030 under RCP4.5, RCP6.0, and RCP8.5, and differences were analyzed relative to a historical period representing the

year 2000. Comparison of simulated temperature and precipitation to reanalysis data showed that the CESM-WRF modeling system performed well for temperature, with absolute biases of less than 2 K for most regions of the U.S. and most seasons of the year. WRF also showed reasonable skill at representing the variability in daily maximum and minimum temperatures throughout the conterminous U.S. Seasonal and annual precipitation biases in the central and eastern U.S. were generally less than 25%, but precipitation was positively biased in the western U.S. throughout the year and in most of the eastern U.S. during

summer.

For the air quality simulations, anthropogenic emissions and boundary conditions were unchanged between the historical and future periods to isolate the meteorological effects of climate change on air quality from non-meteorological factors. Results indicated increases in seasonal mean MDA8 $O_3$ during summer in the Northern Rockies, Upper Midwest, Ohio Valley, and Northeast regions under all scenarios. The increase was largest under RCP8.5, exceeding 4 ppb in parts of the Northern Rockies

and Upper Midwest regions. Smaller increases of up to 2 ppb were simulated during autumn, while changes during spring were generally less than 1 ppb. Increases were magnified at the upper end of the $O_3$ distribution in the Upper Midwest, Ohio Valley, Northeast and Southeast regions. $PM_{2.5}$ concentration changes varied by scenario and by season, with annual average changes of up to ± 1.0 $\mathrm{\mu g\,m^{-3}}$. Decreases in $PM_{2.5}$ were principally due to reductions in aerosol $NO_3^-$ during winter and spring, accompanied by lesser decreases in $NH_4^+$ and $SO_4^{2-}$, due to warmer temperatures causing increased gas-phase partitioning.

Increases in secondary organic aerosol occurred during summer and autumn due to increased biogenic emissions.

Observational evidence (Bloomer et al., 2009) and modeling studies (Rasmussen et al., 2013) have argued that the $O_3$ climate penalty (ppb $K^{-1}$) is lower at reduced levels of $NO_x$ emissions. It is important to recognize that the results presented here use a projected 2030 emission inventory with continued implementation of $NO_x$ emissions controls. The increase in $O_3$ resulting from a given climate scenario would be expected to be greater if $NO_x$ emissions are higher than projected here, particularly in

$NO_x$-limited regions such as the eastern U.S.

The physical and chemical processes that influence air pollutant concentrations are complex, and there are numerous aspects that may potentially vary due to climate change. Quantities examined to account for the modeled changes in pollution included temperature, precipitation, PBL height, wind speed, cloud cover, isoprene emissions, and the number of days having stagnant



weather conditions. Temperature and isoprene emissions were found to have the greatest changes under all scenarios, especially in summer, and the greatest subsequent impact on $O_3$ and $PM_{2.5}$ concentrations.

There are a number of important limitations of the present study. Though biogenic emissions of VOCs were estimated using the downscaled meteorology, our modeling did not consider changes to prevalence and distribution of species of vegetation,
or the potential leaf-scale inhibition of biogenic isoprene emissions due to elevated atmospheric $CO_2$ concentrations (Tai et al., 2013; Sharkey and Monson, 2014). Other natural emissions sources potentially affected by climate change, including wildfires and windblown dust, were neglected in this work. We did not model changes in lightning $NO_x$ formation rates, nor changes in stratosphere-troposphere exchange of $O_3$. We also did not consider changes in drivers of global baseline air pollution, including atmospheric methane levels, foreign emissions scenarios, and long-range transport to the U.S. Finally, there
is substantial interannual variability in air quality due to year-to-year changes in meteorology. Though we conducted four sets of 11-year continuous simulations to account for interannual variability to the extent that our computational resources made practicable, 11-year simulations are likely insufficient to represent the full range of natural variability in the earth's climate system (Garcia-Menendez et al., 2017).

The effects of climate change on $O_3$ and $PM_{2.5}$ obtained in this study are in the range of those reported in similar studies
focused on air quality at 2050 (Fiore et al., 2015, and references therein). However, to our knowledge this study represents the most comprehensive analysis of the potential changes in U.S. regional scale air quality due to climate change conducted to date, in that it encompassed three future climate scenarios for periods exceeding a decade in duration and considered changes in both $O_3$ and $PM_{2.5}$. The significant and widespread increases in model-projected MDA8 $O_3$ associated with specific future climate scenarios, including in some densely populated areas, have potentially important implications for on-going efforts to
reduce exposure to ozone and protect human health.

*Competing interests.* The authors declare that they have no competing interests.

*Disclaimer.* The views expressed in this article are those of the authors and do not necessarily represent the views or policies of the U.S. Environmental Protection Agency.

*Acknowledgements.* CESM (CCSM4) global climate model data corresponding to the 20th century, RCP8.5, RCP6.0, and RCP4.5 "MOAR"
simulations were downloaded from the Earth System Grid via the University Corporation for Atmospheric Research website at www.cesm. ucar.edu/experiments/cesm1.0. The authors thank Lara Reynolds (CSRA) and Daiwen Kang (EPA) for assistance conducting the WRF and CMAQ simulations analyzed here, and Chris Weaver, Darrell Winner, Barron Henderson, and Marcus Sarofim (EPA) for providing critical reviews of a draft of this manuscript.





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



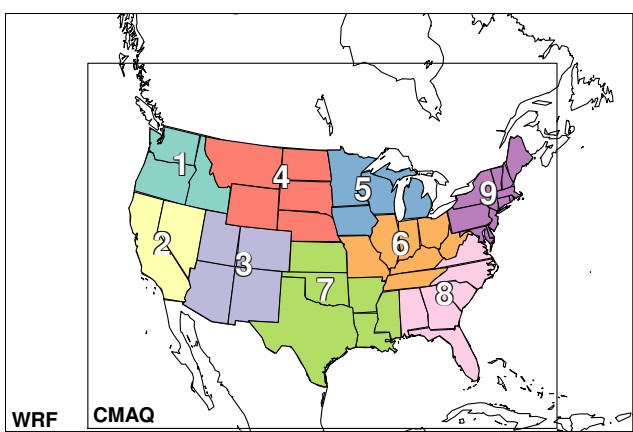

**Figure 1.** WRF and CMAQ modeling domains, with colored areas representing the National Centers for Environmental Information (NCEI) U.S. climate regions used for evaluation: (1) Northwest; (2) West; (3) Southwest; (4) Northern Rockies & Plains; (5) Upper Midwest; (6) Ohio Valley; (7) South; (8) Southeast; and (9) Northeast.



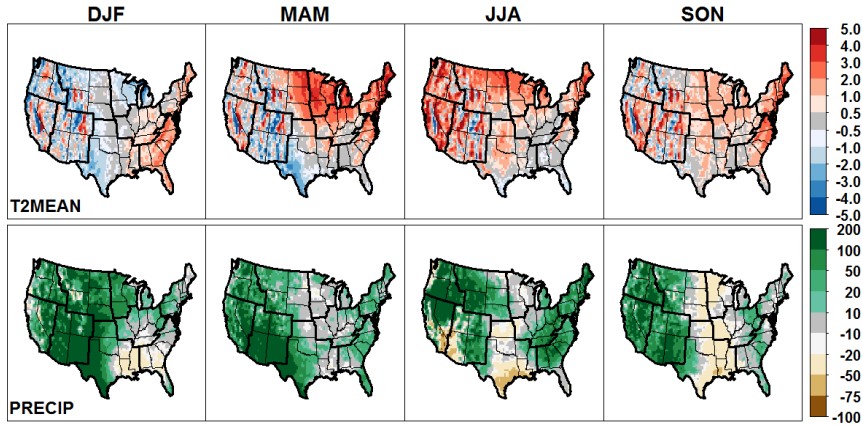

**Figure 2.** Seasonally averaged biases in daily mean 2-m temperature compared to CFSR (K) and precipitation relative to NARR (%) simulated by downscaling CESM with WRF for 1995–2005.


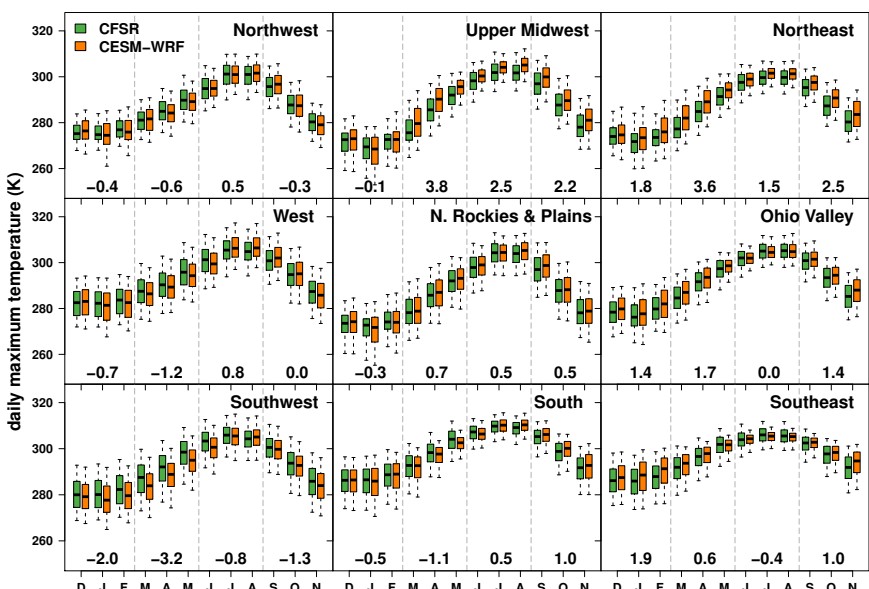

**Figure 3.** Monthly boxplots of daily maximum 2-m temperature simulated by downscaling CESM with WRF for the historical 1995–2005 period compared against CFSR for each of the U.S. climate regions shown in Figure 1. Boxes range from the 25th to 75th percentiles with the dark line denoting the median, and whiskers extend to 5th and 95th percentiles. Seasonal biases (K) are shown at bottom.



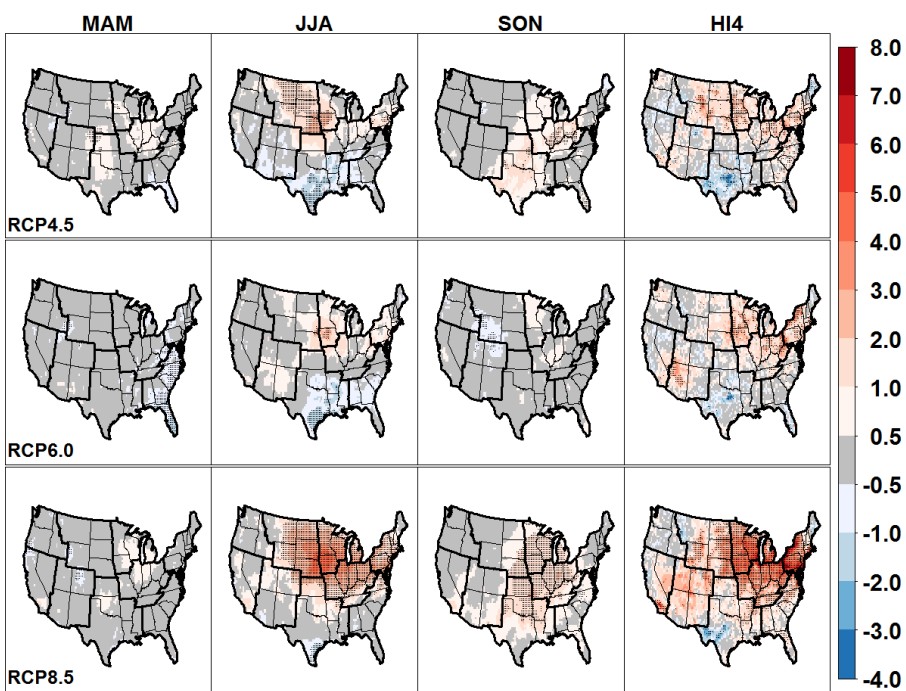

**Figure 4.** Projected changes in maximum daily 8-h average (MDA8) $O_3$ mixing ratios (ppb) from 1995–2005 to 2025–2035 under RCP4.5, RCP6.0, and RCP8.5 (in rows). Columns show projected changes for spring, summer, and autumn seasonal means, as well as 4th-highest annual values ("HI4"). Dark pixels indicate where differences are significant by Student's t-test ($p < 0.05$).



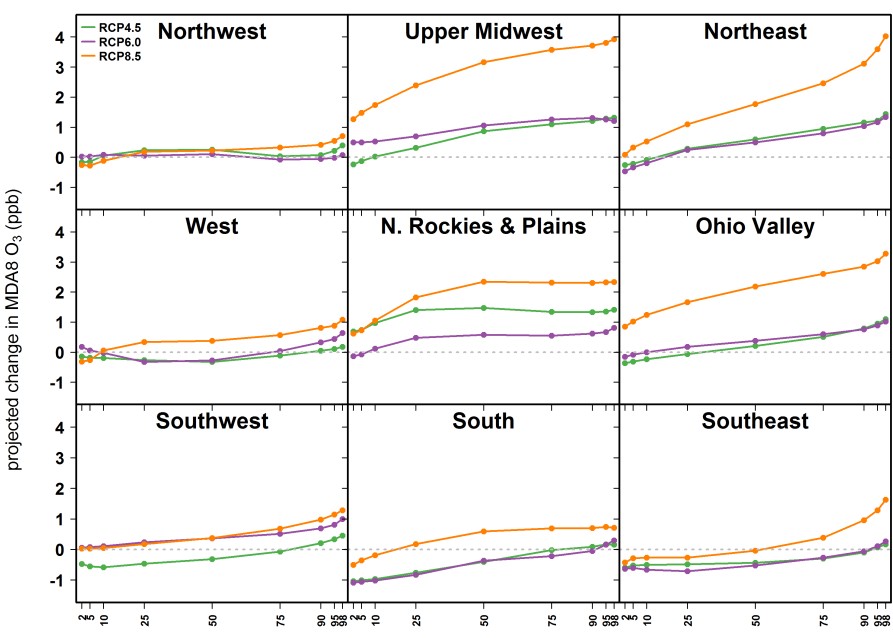

**Figure 5.** Projected changes in percentiles of summer average MDA8 $O_3$ mixing ratios (ppb) simulated by CMAQ under RCP4.5, RCP6.0, and RCP8.5 within each of the U.S. climate regions shown in Fig. 1.



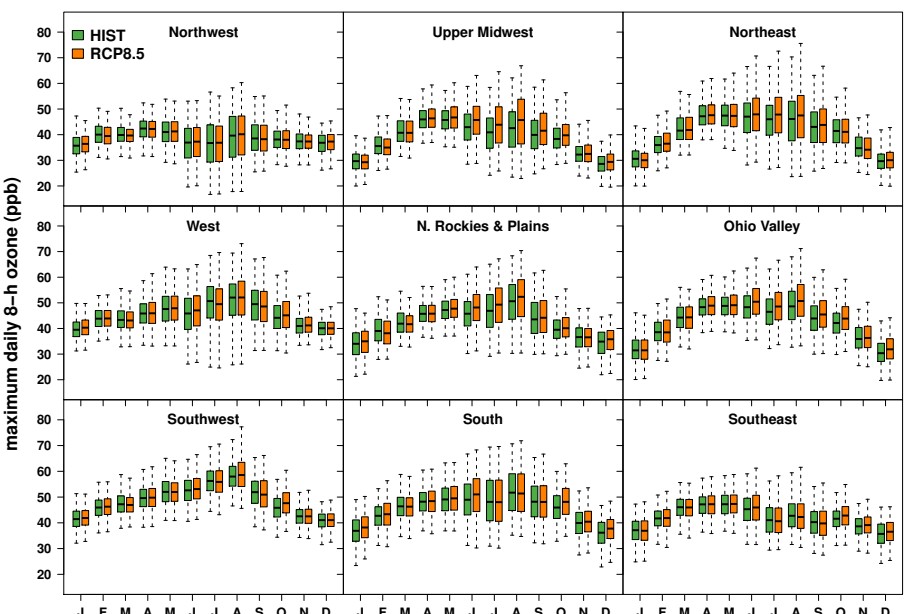

**Figure 6.** Monthly boxplots of MDA8 $O_3$ simulated for the historical 1995-2005 period and 2025–2035 under RCP8.5 for each of the U.S. climate regions shown in Figure 1. Boxes range from the 25th to 75th percentiles with the dark line denoting the median, and whiskers extending to 2nd and 98th percentiles.



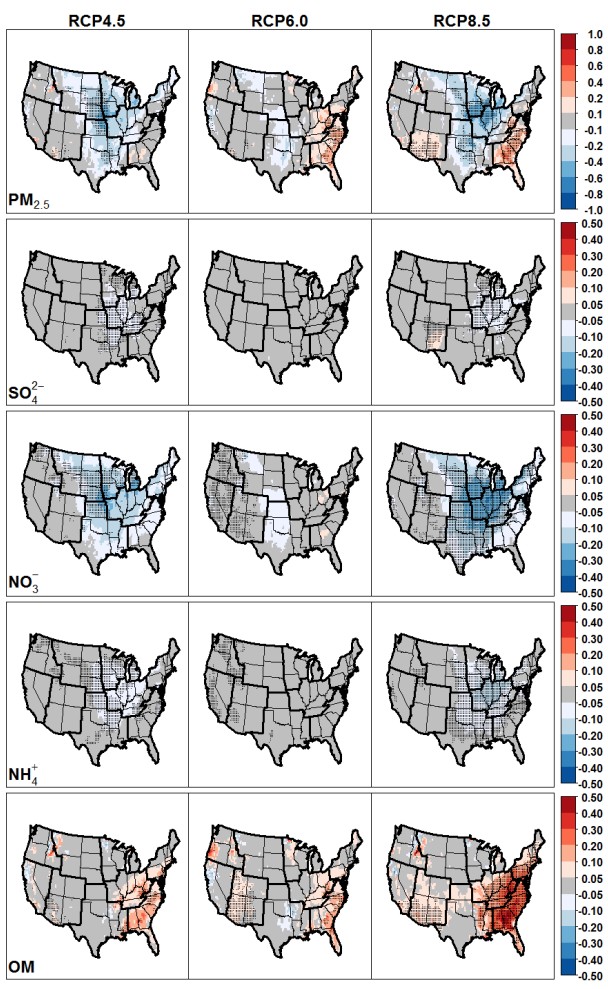

**Figure 7.** Projected changes in annual mean concentrations ($\mu g\,m^{-3}$) of total $PM_{2.5}$ and principal $PM_{2.5}$ components from 1995–2005 to 2025–2035 under RCP4.5, RCP6.0, and RCP8.5. Dark pixels indicate where differences are significant by Student's t-test ($p < 0.05$).





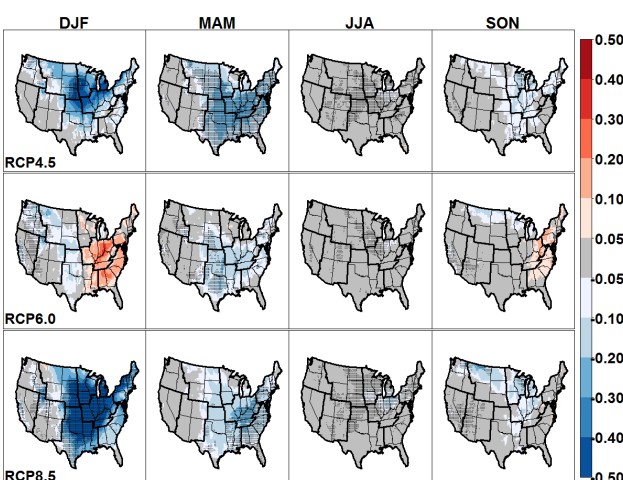

**Figure 8.** Changes in seasonal mean concentrations ($\mu g\,m^{-3}$) of $PM_{2.5}$ nitrate under three RCP scenarios. Dark pixels indicate where differences are significant by Student's t-test ($p < 0.05$).





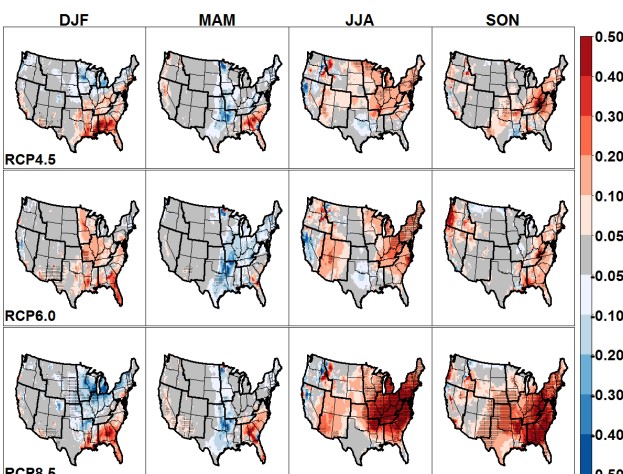

**Figure 9.** Changes in seasonal mean concentrations ($\mu g\,m^{-3}$) of $PM_{2.5}$ organic matter under three RCP scenarios. Dark pixels indicate where differences are significant by Student's t-test ($p < 0.05$).




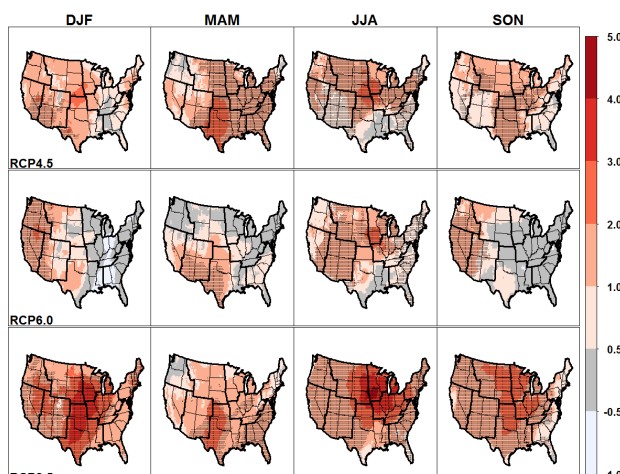

**Figure 10.** Projected changes in seasonal averages of daily maximum 2-m temperature (K) from 1995–2005 to 2025–2035 under RCP4.5, RCP6.0, and RCP8.5. Dark pixels indicate where differences are significant by Student's t-test ($p < 0.05$).




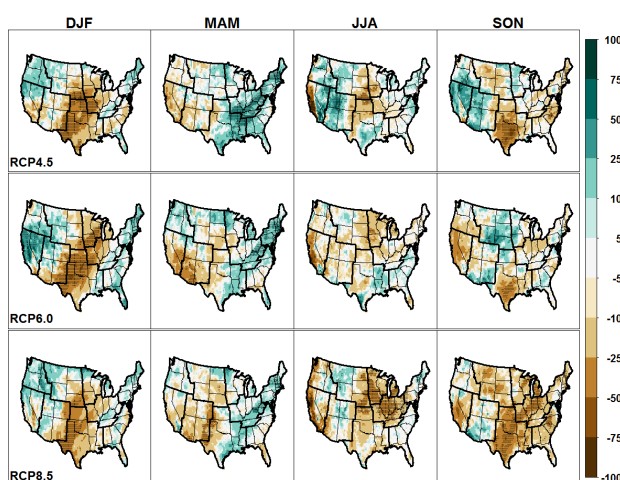

**Figure 11.** Relative changes projected in seasonal accumulated precipitation (%) from 1995–2005 to 2025–2035 under RCP4.5, RCP6.0, and RCP8.5. Dark pixels indicate where differences are significant by Student's t-test ($p < 0.05$).





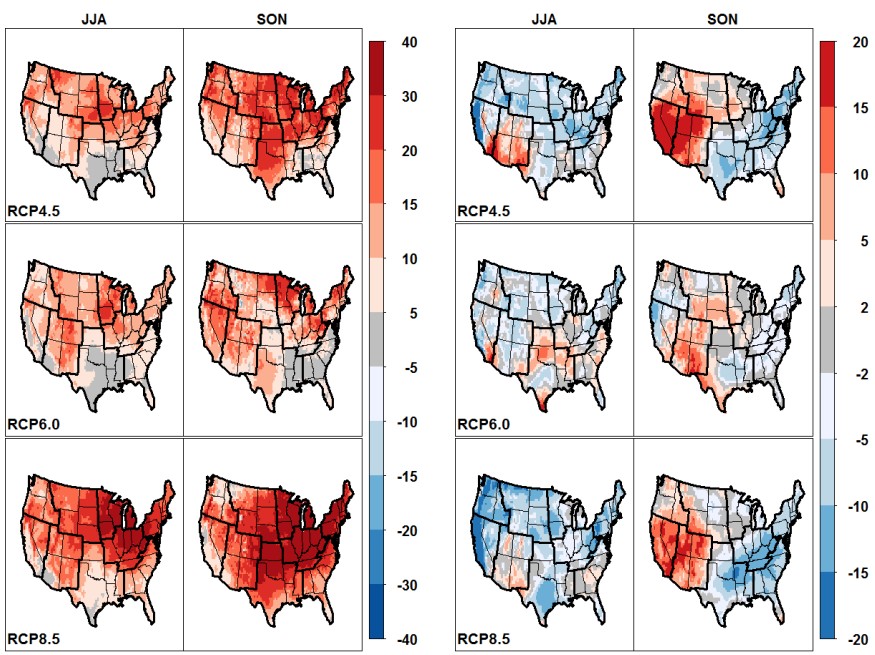

**Figure 12.** Projected changes (percent) in summer and autumn averages of biogenic isoprene emissions (left) and cloud fraction (right) between 1995–2005 and 2025–2035 under RCP4.5, RCP6.0, and RCP8.5.



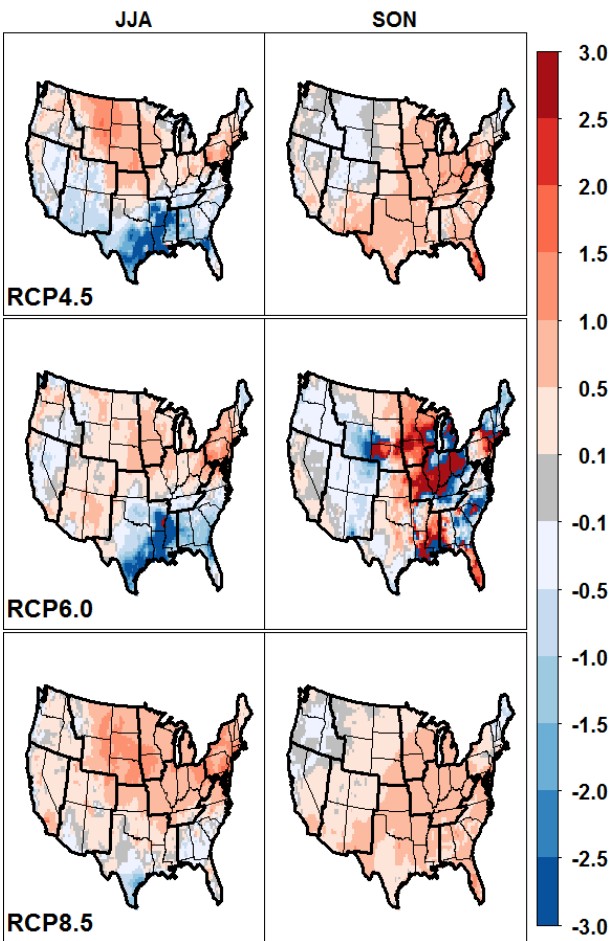

**Figure 13.** Ratio of projected changes in seasonal MDA8 O$_3$ to changes in seasonal daily maximum 2-m temperature (ppb K$^{-1}$) for summer and autumn between 1995–2005 and 2025–2035 under RCP4.5, RCP6.0, and RCP8.5.



**Table 1.** Seasonally and regionally averaged biases in accumulated precipitation (%) in comparison to NARR for 1995–2005

| Region | DJF | MAM | JJA | SON | annual |
|---|---|---|---|---|---|
| Northwest | 34 | 41 | 118 | 51 | 48 |
| West | 27 | 70 | 151 | 80 | 54 |
| Southwest | 175 | 132 | 41 | 80 | 97 |
| N. Rockies | 66 | 20 | 53 | 13 | 36 |
| Upper Midwest | 35 | −1 | 1 | −16 | 0 |
| Ohio Valley | 5 | 4 | 33 | −3 | 11 |
| South | 17 | 25 | −17 | −20 | 0 |
| Southeast | −5 | 23 | 41 | 13 | 20 |
| Northeast | 14 | 23 | 65 | 7 | 28 |



**Table 2.** Changes in maximum daily 8-h $O_3$ by region

| Region | RCP4.5 | | | | RCP6.0 | | | | RCP8.5 | | | |
|---|---|---|---|---|---|---|---|---|---|---|---|---|
| | MAM | JJA | SON | HI4 | MAM | JJA | SON | HI4 | MAM | JJA | SON | HI4 |
| Northwest | −0.1 | 0.2 | −0.1 | 0.1 | −0.2 | 0.0 | −0.3 | 0.3 | −0.3 | 0.2 | −0.1 | 0.0 |
| West | −0.1 | −0.2 | 0.1 | 0.1 | −0.1 | −0.1 | 0.1 | 0.2 | 0.0 | 0.4 | 0.3 | 1.1 |
| Southwest | 0.2 | −0.2 | 0.2 | 0.4 | 0.1 | 0.4 | −0.2 | 0.6 | −0.1 | 0.5 | 0.5 | 1.6 |
| N. Rockies | 0.1 | 1.3 | 0.0 | 1.1 | −0.2 | 0.5 | −0.1 | 0.6 | −0.0 | 2.0 | 0.5 | 1.6 |
| Upper Midwest | 0.4 | 0.7 | 0.6 | 1.4 | −0.3 | 1.0 | 0.5 | 1.7 | 0.4 | 2.9 | 1.3 | 3.8 |
| Ohio Valley | 0.5 | 0.2 | 0.9 | 1.0 | −0.3 | 0.4 | 0.4 | 1.2 | 0.3 | 2.1 | 1.4 | 3.1 |
| South | 0.2 | −0.4 | 0.8 | −0.5 | −0.1 | −0.5 | 0.0 | −0.1 | 0.1 | 0.4 | 1.0 | 0.6 |
| Southeast | −0.1 | −0.4 | 0.3 | 0.7 | −0.7 | −0.4 | 0.1 | 0.2 | −0.0 | 0.2 | 0.5 | 1.2 |
| Northeast | 0.2 | 0.6 | 0.1 | 1.1 | −0.4 | 0.5 | −0.2 | 1.5 | 0.2 | 1.8 | 0.2 | 3.3 |
| all US | 0.2 | 0.2 | 0.4 | 0.5 | −0.2 | 0.1 | 0.0 | 0.6 | 0.0 | 1.1 | 0.7 | 1.6 |