# Peer review of "The potential effects of climate change on air quality across the conterminous U.S. at 2030 under three Representative Concentration Pathways"

_Atmospheric Chemistry and Physics, 2018_

## Referee Comment (RC1) · Anonymous Referee #4 · 4 Jul 2018

This paper describes a multi decade air quality simulation over the contiguous US using a regional scale application of the model "CMAQ" with downscaled meteorology from global climate scenarios. The authors employ constant anthropogenic emissions and investigate changes to ambient concentrations of ozone and PM2.5 (mass and chemical composition) due to temperature/climate changes only. They find the largest changes in [PM2.5] come from reduction in [NO3] in winter and increased [OM] in warmer seasons, presumably due to higher biogenic VOC emissions. Though the authors provide no direct evidence for the biogenic emission/higher [OM] - but I do

happen to agree. The authors find the largest changes in 8 hour max ozone occur at the higher end of the distribution. Their findings are consistent with many previous studies. The downscaling and model applications methods are done well. The figures in the manuscript and supplemental information are excellent. However the analysis is not as strong as the rest of this work. Below I list specific reasons why I think this.

This paper contributes to the body of knowledge indicating temperature and air quality relationships. The authors have a unique opportunity to evaluate chemical-temperature trends and better context is needed.

Specific Comments: A powerful motivation for this study is that future PM2.5 is less well constrained than ozone. The authors use of CMAQ with detailed particle chemistry in long term simulations is an improvement over global scale models (with less particle chemistry) typically used in such research.

The model simulations have been conducted for time periods for which changes in ambient values of O3 and PM2.5 have been recorded. Links of measurements to EPA policy and temperature change can be evaluated. Why is this not part of the model evaluation and work presented in this manuscript? If the authors expect confidence in the future relationships they present, evaluation of past trends and relationships for retrospective periods builds confidence for their assessment and is necessary. What are the current d[O3]/dT and d[PM2.5]/dT, for example in the regions outlined in Figure 1? Can they be replicated by the modeling system?

There are temperature dependent anthropogenic emissions. Electricity sector emissions, in particular in the United States (e.g., California: Miller et al., 2008; Farkas et al., 2016), change with increasing temperature and this is not captured in this work. The absence of such relationships suggest changes at peak O3 and PM2.5 pollution is under represented here. The authors should note this and explain the uncertainty, the complications this introduces, in particular when they describe changes at the peak end of pollutant distributions.
Starting at Line 22, page 1: The authors state that due partly to Tier 3 emission standards for motor vehicles, anthropogenic emissions are expected to decrease through 2030. Is this still true? How does the Ozone Standards Implementation Act of 2017 affect/not affect expected trends in emissions and ambient air quality?

The references for AERO6 (Simon and Bhave, Nolte et al.) are insufficient to describe the AERO6 module. Further, the authors discuss that some of the largest [PM] prediction changes are due to temperature induced changes on BVOC emissions that affect [OM] predictions in some portions of the the US. The chosen references do not explain why this would be the case in the model at all. Please provide better reference(s) that help readers understand the relationship between biogenic VOC emissions and the connection it PM2.5 OM (presumably biogenic secondary organic aerosol) in CMAQ.

Line 25, Page 5: Can the authors explain what "wet bias" means and the implications? Does this mean excess precipitation? Does this mean the implication is there is more wash out/cleaning of the atmosphere?

Line 12, Page 9: "This supports the conclusion that warmer temperatures in a future climate results in increased partitioning of aerosol NO3 to HNO3" Presumably, the authors can test this idea/hypothesis in their model output?

Editorial

Line 22: "Pope III", is that formatting correct?

References: Farkas et al., "High electricity demand in the northeast U.S.: PJM reliability network and peaking unit impacts on air quality", Environ. Sci. Technol. 2016, 50(15), pp 8375-8384. Miller et al., "Climate, extreme heat, and electricity demand in California", J. App. Met. Clim. 2008, 47, pp. 1834-1844.

---

## Referee Comment (RC2) · Anonymous Referee #2 · 10 Jul 2018

Nolte et al. estimate the impact of climate change on U.S. air quality by using a pipeline of models (CESM->WRF->CMAQ). The results are not particularly novel, but the method employed by the authors is a step forward in refining estimates of the effects of climate change on air quality. The results are an important addition to the literature. The authors find that impacts on ozone and PM have important regional and seasonal subtleties, but generally reveal an increase in ozone, decrease in nitrate, and increase in organic matter. The manuscript is well-written and presented very clearly. I recommend publication in ACP following sufficient response to the following minor

comments.

General Comments - The authors spend a lot of real estate discussing model biases in temperature and precipitation. The principal source of bias in Nolte et al. (2008) was temperature, but what about other factors? After all, part of temperature's explanatory power arises from its ability to be a catch-all for many factors. In the latter half of the paper other factors are revealed to be important (related to T), including cloud cover, isoprene emissions, and stagnation (circulation). If evaluation of air pollution meteorology is important, these additional factors must surely be of interest.

- The authors assume the results of the model are truth, but indeed the model is programed with the assumptions that HNO3 is less soluble at higher temperatures (well known to be so, but what is the sensitivity?) and that isoprene emissions increase with temperature. But are the sensitivities of these factors to temperature accurate? This seems more important than any absolute bias in temperature, the changes in meteorology and air quality are of greatest interest here. At the very least, more discussion of parameters/observational evidence underscoring the principal impacts is necessary, e.g., change in isoprene emissions.

- 11-years is still potentially too short to average out interannual variability and obtain a robust climate signal. It can take decades and an ensemble to do that. I think that this manuscript is a step in the right direction in terms of incorporating multiple climate scenarios and a longer record, but it still need to acknowledge that interannual variability can still distort results.

Specific Comments Title: Perhaps a nitpick, but the parenthetical seems unnecessary in a title

Abstract, line 10-12: It might be worth referencing the changes here to be driven by climate change only. It is a little confusing since the emission scenario is mentioned in reference to GHGs and not O3/PM precursors

Page 4, line 29 - page 5, line 2: Is this description of emission changes related to the lateral boundary condition simulation? If not, this should be reworded to make this clearer given its following of the discussion about the boundary conditions.

Page 5, final paragraph: Is the discussion of max/min temperatures really necessary here since it is rehashed in depth in the following paragraph. This was a bit jarring on the first read.

Page 6: Why are these evaluations important? There should be some discussion here about what a bias in temperature and/or precipitation means for the present study. How much is gained by evaluating the maximum and minimum temperatures, in addition to the daily mean? This goes along with the second bullet in the General Comments section.

Page 6, Line 31: Be a little careful here because the spatial similarities could arise from the common baseline, which is subtracted from each simulation.

Page 8, Line 13-14: Why are sulfate and ammonium decreasing?

Page 9, Line 12: This doesn't really support the conclusion since the model is programmed this way. Only observational evidence would really support the conclusion.

---

## Author Comment (AC1) · 16 Oct 2018

The referee's comments are noted in italics below, followed by our responses.

*This paper describes a multi decade air quality simulation over the contiguous US using a regional scale application of the model "CMAQ" with downscaled meteorology from global climate scenarios. The authors employ constant anthropogenic emissions and investigate changes to ambient concentrations of ozone and PM2.5 (mass and chemical composition) due to temperature/climate changes only. They find the largest changes in [PM2.5] come from reduction in [NO3] in winter and increased [OM] in warmer seasons, presumably due to higher biogenic VOC emissions. Though the authors provide no direct evidence for the biogenic emission/higher [OM] - but I do happen to agree. The authors find the largest changes in 8 hour max ozone occur at the higher end of the distribution. Their findings are consistent with many previous studies. The downscaling and model applications methods are done well. The figures in the manuscript and supplemental information are excellent. However the analysis is not as strong as the rest of this work. Below I list specific reasons why I think this.*

*This paper contributes to the body of knowledge indicating temperature and air quality relationships. The authors have a unique opportunity to evaluate chemical-temperature trends and better context is needed.*

We thank the referee for the constructive comments on our manuscript.

*Specific Comments: A powerful motivation for this study is that future PM2.5 is less well constrained than ozone. The authors use of CMAQ with detailed particle chemistry in long term simulations is an improvement over global scale models (with less particle chemistry) typically used in such research. The model simulations have been conducted for time periods for which changes in ambient values of O3 and PM2.5 have been recorded. Links of measurements to EPA policy and temperature change can be evaluated. Why is this not part of the model evaluation and work presented in this manuscript? If the authors expect confidence in the future relationships they present, evaluation of past trends and relationships for retrospective periods builds confidence for their assessment and is necessary. What are the current d[O3]/dT and d[PM2.5]/dT, for example in the regions outlined in Figure 1? Can they be replicated by the modeling system?*

CMAQ has been extensively evaluated as a chemical transport model, including "dynamic evaluation" of changes in simulated ozone levels in response to changes in emissions and meteorology (e.g., Gilliland et al., 2008; Foley et al., 2015). The simulations conducted in this study used meteorology downscaled from global climate model (CESM) simulations of the historical period 1995-2005 and of the future period 2025-2035 under three scenarios of greenhouse gas trajectories and radiative forcing. Because the effect of air pollutant (principally $NO_x$ and $SO_2$, but also VOC) emissions changes on air quality is much larger than the effect of climate change-driven changes in meteorology over this period, we used constant levels of anthropogenic emissions in all our CMAQ simulations. This enables us to estimate quantitatively the impact on air pollutant concentrations of climate change in isolation from other factors, such as changes in domestic and international emissions of air pollutants. However, it would be inappropriate to evaluate ozone and PM concentrations from the CMAQ simulations in this study against historical measurements, because the emissions used in this study represented projections of future conditions, and were much lower than actual historical emissions. The $d[O_3]/dT$ and $d[PM_{2.5}]/dT$ modeled under such a different emissions regime would not be directly comparable to $d[O_3]/dT$ and $d[PM_{2.5}]/dT$ based on historical observations. This point is discussed in the Conclusions section (bottom of p. 10):

> **Observational evidence (Bloomer et al., 2009) and modeling studies (Rasmussen et al., 2013) have argued that the $O_3$ climate penalty (ppb $K^{-1}$) is lower at reduced levels of $NO_x$ emissions. It is important to recognize that the results presented here use a projected 2030 emission inventory with continued implementation of $NO_x$**

**emissions controls. The increase in O₃ resulting from a given climate scenario would be expected to be greater if NOₓ emissions are higher than projected here, particularly in NOₓ-limited regions such as the eastern U.S.**

Instead, we have taken an approach in which we first evaluated CMAQ using downscaled historical meteorology and historical emissions changes in comparison to measurements of air pollutant concentrations over the period 2000-2010 (Seltzer et al., 2016). That study demonstrated model performance using downscaled meteorology was comparable to that typically obtained in standard air quality applications, which provides confidence in the overall method employed. Of course, regional air quality model results obtained using this methodology depend critically on the global climate model simulation being downscaled.

References:
Gilliland, A.B., C. Hogrefe, R.W. Pinder, J.M. Godowitch, K.L. Foley, S.T. Rao (2008): Dynamic evaluation of regional air quality models: Assessing changes in O3 stemming from changes in emissions and meteorology. *Atmos. Environ.* 42, 5110-5123.

Foley, K.M., C. Hogrefe, G. Pouliot, N. Possiel, S.J. Roselle, H. Simon, B. Timon (2015): Dynamic evaluation of CMAQ part I: Separating the effects of changing emissions and changing meteorology on ozone levels between 2002 and 2005 in the eastern US. *Atmos. Environ.* 103, 247-255.

*There are temperature dependent anthropogenic emissions. Electricity sector emissions, in particular in the United States (e.g., California: Miller et al., 2008; Farkas et al., 2016), change with increasing temperature and this is not captured in this work. The absence of such relationships suggest changes at peak O3 and PM2.5 pollution is under represented here. The authors should note this and explain the uncertainty, the complications this introduces, in particular when they describe changes at the peak end of pollutant distributions.*

We have added this point to the paragraph in the Conclusions (p. 11) discussing limitations of the present study:

**To isolate the effect of climate change on air quality, we kept anthropogenic emissions constant across all modeled years. However, electric sector emissions increase during peak temperature events due to increased demand for air conditioning, and emissions from electric generating units used to provide power during peak periods are less strictly regulated (Farkas et al., 2016). The increased emissions associated with increased electricity demand during heat waves is not represented in our analysis, potentially underestimating the impact on upper percentile and annual 4th-highest O₃ levels.**

*Starting at Line 22, page 1: The authors state that due partly to Tier 3 emission standards for motor vehicles, anthropogenic emissions are expected to decrease through 2030. Is this still true? How does the Ozone Standards Implementation Act of 2017 affect/not affect expected trends in emissions and ambient air quality?*

Our statement and analysis is based on existing legislation and regulations. The Ozone Standards Implementation Act of 2017 is a bill that has passed the U.S. House of Representatives but has not yet been acted upon by the Senate. Accordingly, it does not yet have the force of law.

*The references for AERO6 (Simon and Bhave, Nolte et al.) are insufficient to describe the AERO6 module. Further, the authors discuss that some of the largest [PM] prediction changes are due to temperature induced changes on BVOC emissions that affect [OM] predictions in some portions of the the US. The chosen references do not explain why this would be the case in the model at all. Please provide better reference(s) that help readers understand the relationship between biogenic VOC emissions and the connection it PM2.5 OM (presumably biogenic secondary organic aerosol) in CMAQ.*

We have added a reference to Carlton et al. (2010), which describes the secondary organic aerosol model in this version of CMAQ. Additionally, we now cite Carlton and Baker (2011) for the BEIS biogenic emissions module.

*Line 25, Page 5: Can the authors explain what "wet bias" means and the implications? Does this mean excess precipitation? Does this mean the implication is there is more wash out/cleaning of the atmosphere?*

We have modified the text to read "while CFSR precipitation is positively biased."  As noted in the Discussion section (p. 9), "Scavenging of soluble aerosols by precipitation is an important removal process for atmospheric particulate matter."  Overestimated precipitation (especially too-frequent precipitation) would overestimate removal of particulate matter from the atmosphere due to scavenging and wet deposition, and therefore result in underestimated $PM_{2.5}$ concentrations.

*Line 12, Page 9: "This supports the conclusion that warmer temperatures in a future climate results in increased partitioning of aerosol NO3 to HNO3" Presumably, the authors can test this idea/hypothesis in their model output?*

This sentence has been deleted. We examined changes in seasonal mean concentrations of gas-phase $HNO_3$ as well as the fraction of total nitrate in the gas phase ($[HNO_3]/[TNO_3]$). Both showed some increases in areas where aerosol $NO_3$ decreased during winter, but the changes in $HNO_3$ and $HNO_3/TNO_3$ were smaller and less widespread than the change in $NO_3$.

*Editorial*
*Line 22: "Pope III", is that formatting correct?*

We have modified the text so that the parenthetical citation is (Pope, 2007) and the full reference reads Pope, III, C.A.

---

## Author Comment (AC2) · 16 Oct 2018

The referee's comments are noted in italics below, followed by our responses.

*Nolte et al. estimate the impact of climate change on U.S. air quality by using a pipeline of models (CESM->WRF->CMAQ). The results are not particularly novel, but the method employed by the authors is a step forward in refining estimates of the effects of climate change on air quality. The results are an important addition to the literature. The authors find that impacts on ozone and PM have important regional and seasonal subtleties, but generally reveal an increase in ozone, decrease in nitrate, and increase in organic matter. The manuscript is well-written and presented very clearly. I recommend publication in ACP following sufficient response to the following minor comments.*

We thank the referee for these comments.

*General Comments - The authors spend a lot of real estate discussing model biases in temperature and precipitation. The principal source of bias in Nolte et al. (2008) was temperature, but what about other factors? After all, part of temperature's explanatory power arises from its ability to be a catch-all for many factors. In the latter half of the paper other factors are revealed to be important (related to T), including cloud cover, isoprene emissions, and stagnation (circulation). If evaluation of air pollution meteorology is important, these additional factors must surely be of interest.*

We agree with the referee that those other factors are important for air pollution meteorology, which is why we examine them in this paper. We discuss Nolte et al. (2008) to place the current manuscript in the proper context with prior work, but an evaluation of the "current" climate variables in that study is outside the scope of the present manuscript. Here, we focus our analysis of historical (1995-2005) climate fields on temperature and precipitation, the two most important and most commonly evaluated variables in regional climate modeling studies. As the reviewer notes, temperature is important due to its relationships with other variables. Rather than evaluate mean values for additional climate variables, we chose to investigate temporal and spatial variability in order to understand the implications of these changes for future air quality.

*- The authors assume the results of the model are truth, but indeed the model is programed with the assumptions that HNO3 is less soluble at higher temperatures (well known to be so, but what is the sensitivity?) and that isoprene emissions increase with temperature. But are the sensitivities of these factors to temperature accurate? This seems more important than any absolute bias in temperature, the changes in meteorology and air quality are of greatest interest here. At the very least, more discussion of parameters/observational evidence underscoring the principal impacts is necessary, e.g., change in isoprene emissions.*

We agree that there is uncertainty in how isoprene and other biogenic VOC emissions will change in response to elevated temperatures and $CO_2$. The Conclusions section contains a paragraph discussing some caveats and limitations of our study, which includes a discussion of isoprene:
**"Although biogenic emissions of VOCs were estimated using the downscaled meteorology, our modeling did not consider changes to prevalence and distribution of species of vegetation, or the potential leaf-scale inhibition of biogenic isoprene emissions due to elevated atmospheric $CO_2$ concentrations (Tai et al., 2013; Sharkey et al., 2014)."**

*- 11-years is still potentially too short to average out interannual variability and obtain a robust climate signal. It can take decades and an ensemble to do that. I think that this manuscript is a step in the right direction in terms of incorporating multiple climate scenarios and a longer record, but it still need to acknowledge that interannual variability can still distort results.*

We agree with the reviewer's concerns and have expressed this within the last two sentences of the second-to-last paragraph:

> **Finally, there is substantial interannual variability in air quality due to year-to-year changes in meteorology. Though we conducted four sets of 11-year continuous simulations to account for interannual variability to the extent that our computational resources made practicable, 11-year simulations are insufficient to represent the full range of natural variability in the earth's climate system (Garcia-Menendez et al., 2017).**

*Specific Comments*

*Title: Perhaps a nitpick, but the parenthetical seems unnecessary in a title.*

We have removed the parenthetical "(RCPs)" from the title of the manuscript.

*Abstract, line 10-12: It might be worth referencing the changes here to be driven by climate change only. It is a little confusing since the emission scenario is mentioned in reference to GHGs and not O3/PM precursors*

The abstract has been revised as suggested by the reviewer:

> **The analysis isolates the future air quality differences arising from climate-driven changes in meteorological parameters and specific natural emissions sources that are strongly influenced by meteorology. Other factors that will affect future air quality, such as anthropogenic air pollutant emissions and chemical boundary conditions, are unchanged across the simulations.**

*Page 4, line 29 - page 5, line 2: Is this description of emission changes related to the lateral boundary condition simulation? If not, this should be reworded to make this clearer given its following of the discussion about the boundary conditions.*

The last sentence of the previous paragraph reads "All other input variables, including anthropogenic emissions, chemical lateral boundary conditions, and land use and land cover classifications, were unchanged across the air quality modeling scenarios." We have restructured this paragraph to place the discussion of domestic emissions prior to the discussion of lateral boundary conditions, in the same order as mentioned above. We additionally specify that the emissions referred to in this paragraph are "air pollutant" emissions.

*Page 5, final paragraph: Is the discussion of max/min temperatures really necessary here since it is rehashed in depth in the following paragraph. This was a bit jarring on the first read.*

As noted in the previous paragraph, the evaluation of the historical period is focused on (1) **monthly and seasonal means** and (2) **selected percentiles** of regional temperature and precipitation. The paragraph at the end of page 5 and Figure 2 pertain to seasonal means. The following paragraph and Figure 3 discuss percentiles of daily maximum temperatures for different regions of the U.S. We feel that analysis of the distribution of temperatures (and ozone concentrations) is a strength of our paper, which goes beyond the use of mean temperatures as has been done in most prior studies.

*Page 6: Why are these evaluations important? There should be some discussion here about what a bias in temperature and/or precipitation means for the present study. How much is gained by evaluating the maximum and minimum temperatures, in addition to the daily mean? This goes along with the second bullet in the General Comments section.*

We analyze daily maximum and daily minimum temperatures, along with precipitation, to evaluate how well the model is representing the climate during the historical period. As stated above, we feel that analysis of distributions of temperature and ozone is a strength of this paper. Good model performance for the historical period provides some degree of confidence in the use of the model for future climate projections. In addition, MDA8 O3 concentrations are often more strongly correlated with daily maximum temperatures than with daily mean temperatures.

*Page 6, Line 31: Be a little careful here because the spatial similarities could arise from the common baseline, which is subtracted from each simulation.*

Here we show the changes in seasonal mean MDA8 O3 for each of the RCPs at 2025-2035 relative to the 1995-2005 baseline. We agree with the Referee that plots of changes are all affected by the common baseline, but we do not see how that undermines the point we are making, which is that the locations of the seasonal changes are generally consistent across the three scenarios.

*Page 8, Line 13-14: Why are sulfate and ammonium decreasing?*

As shown in Fig. S8 of the SI, the changes in sulfate are small (less than 0.3 $\mu$g m$^{-3}$), are mixed in sign, and vary by season and scenario, likely due to a complex combination of factors including chemical formation rates and changes in transport. The changes in ammonium are somewhat larger and more clearly track the changes in nitrate to maintain thermodynamic equilibrium.

*Page 9, Line 12: This doesn't really support the conclusion since the model is programmed this way. Only observational evidence would really support the conclusion.*

This sentence has been deleted.